# Hemp Fibre Properties and Processing Target Textile: A Review

**DOI:** 10.3390/ma15051901

**Published:** 2022-03-03

**Authors:** Malgorzata Zimniewska

**Affiliations:** Institute of Natural Fibres and Medicinal Plants—National Research Institute, Wojska Polskiego 71b, 60-630 Poznan, Poland; malgorzata.zimniewska@iwnirz.pl; Tel.: +48-61-8455-8830

**Keywords:** hemp fibre, fibre structure, hemp retting, spinning, value chain

## Abstract

Over the last several decades, *Cannabis sativa* L. has become one of the most fashionable plants. To use the hemp potential for the development of a sustainable textile bio-product sector, it is necessary to learn about the effect of the processes creating hemp’s value chain on fibre properties. This review presents a multi-perspective approach to industrial hemp as a resource delivering textile fibres. This article extensively explores the current development of hemp fibre processes including methods of fibre extraction and processing and comprehensive fibre characteristics to indicate the challenges and opportunities regarding *Cannabis sativa* L. Presented statistics prove the increasing interest worldwide in hemp raw material and hemp-based bio-products. This article discusses the most relevant findings in terms of the effect of the retting processes on the composition of chemical fibres resulting in specific fibre properties. Methods of fibre extraction include dew retting, water retting, osmotic degumming, enzymatic retting, steam explosion and mechanical decortication to decompose pectin, lignin and hemicellulose to remove them from the stem with varying efficiency. This determines further processes and proves the diversity of ways to produce yarn by employing different spinning systems such as linen spinning, cotton and wool spinning technology with or without the use of the decortication process. The aim of this study is to provide knowledge for better understanding of the textile aspects of hemp fibres and their relationship to applied technological processes.

## 1. Introduction

Over the last several decades, *Cannabis sativa* L. has become one of the most fashionable plants. The huge ecological potential of the plants, and the diversity of raw materials that can be delivered by the plant, makes industrial hemp interesting for agriculture, medicine, food, textiles, construction and other industries. Wide recognition of the medical value of cannabidiol CBD [1,2,3] has led farmers to focus on *Cannabis sativa* L. cultivation targeted to CBD extract production. The interest in CBD use is reflected in the size of the global cannabidiol market, which was valued at USD 2.8 billion in 2020 and is expected to expand at a compound annual growth rate (CAGR) of 21.2% from 2021 to 2028 [4].

Growing appreciation for hemp is due to its suitability for helping to address high necessity for environmental protection due to increasing air temperature and related to this, climatic changes. The development of the European Green Deal (EGD) determines strategy focused on the modern transformation of the European Union in terms of resource efficiency and competitive economies with particular emphasis on a holistic approach toward climate and environmental challenges [5].

The principles of EGD and other European strategies like the Circular Economy Action Plan, Farm to Fork Strategy, the EU Climate Pact and others recommend the reduction of greenhouse gasses emissions, implementation of a circular economy, effective natural resource management and the replacement of fossil fuels with renewable energy resources. Hemp is an annual plant characterised by a well-developed leaf system and is one of the fastest growing plants on Earth [6]. It can absorb approximately 10 t of CO_2_ (depending on plant variety) from the atmosphere during one vegetation period, improving air quality, thermal balance and ensuring a positive environmental impact [7].

The additional benefits of hemp growing are the suppression of weed growth, anti-erosion, reclamation properties and the ability to drain the soil of poisonous substances and heavy metals [8]. The ability of hemp plants to kill tough weeds results from several factors such as the tall height of the hemp plants, its thick leaves, and fact that it can be densely cultivated [9].

Farmers growing hemp can reduce the amount of herbicides used, implement crop rotation and gain organic certification. All the environmental aspects of hemp cultivation and the growing hemp product market has made the plant attractive for the agricultural sector.

All the stages of the total value chain of *Cannabis sativa* L. e.g., growing, processing, use and finally recycle/reuse/bio-refination/waste management, fulfil the principles of the above-mentioned strategies and can contribute to the fight against climate change.

Hemp has huge potential for multidirectional applications. The holistic approach to hemp from the agricultural sector, industry, consumers, circular economy and environmental perspective positions hemp as one of the most important plants within the bio-economy suitable to grow to address future perspectives and initiatives.

All the arguments listed above have resulted in rising global interest in hemp cultivation and use, as well as predictions about the growing hemp market size in the future. Figure 1 shows the forecast picture for the U.S. industrial market size during the period from 2016–2027.

The global market size estimated for industrial hemp was USD 4.71 billion in 2019. It is expected that the annual growth rate will be 15.8% over the forecast period. This situation results from increasing interest in hemp-based products such as seeds for oil, food and beverages as well as fibres used for technical applications, for example, as a composite used mainly for automotive and construction purposes, but also in the textile sector, especially in emerging regions such as the Asia Pacific [10].

The hemp industry is also growing rapidly in Europe, where the full potential of hemp is not even fully used. The area of hemp cultivation in Europe increased by 70% from 2013 to 2018. In comparison to 1993 figures, the number of hectares has increased 614% since [11].

The whole of the hemp has the potential to be used, which delivers raw materials for many sectors of economy. The possible directions for the use of the hemp plant are presented in Figure 2. Figure 3 shows hemp-based product sales by category in 2015.

Both Figure 2 and Figure 3 prove the importance of hemp fibre use in bio-product sectors. The hemp fibre market illustrated in Figure 4 was valued at USD 4.46 Billion in 2019 and is projected to reach USD 43.75 Billion by 2027, growing at a CAGR of 33% from 2020 to 2027 [14].

Due to huge market demand, stakeholders are looking for hemp fibre technologies suitable to implement in current conditions, when a lack of machines necessary for fibre processing is a common problem in Europe and other macro-regions throughout the world [11].

To respond to industry needs, many researchers are addressing the hemp problems and are working on developing new technological solutions.

The aim of this paper is to present the multi-perspective approach to industrial hemp as a resource delivering valuable fibre and provide comprehensive knowledge based on reviews of the available literature including the newest findings, which describe hemp fibre properties, methods of extraction and processing to highlight the challenges and opportunities concerning the fibre obtained from *Cannabis sativa* L.

## 2. Hemp Fibres Characteristics

### 2.1. Hemp Stalk and Fibre Structure

The stalk of a matured hemp plant is made up of several layers, which is illustrated in Figure 5. The cuticle/epidermis is the outside layer which protects the cells of the stalk against moisture evaporation and sudden temperature changes, as well as partly giving mechanical reinforcement to the stem. The epidermis consists of colour-binding media and stomata (pores), through which the plant ventilates and regulates evaporation [14].

The thin cortex containing chlorophyll adheres to the phloem layer with bundles of bast fibres. Xylem, pith and cambium create the large woody layer which is approximately 75% of the total stalk mass and is responsible for the transportation of the soluble organic compounds created during photosynthesis as well as water and mineral distribution from the roots to the whole plant. Xylem, which is in the middle part of the plant, consists of parenchyma and vessels, which both have transport functions, and libriform fibres (core fibres), which give the plant rigidity and strength [16].

Fibre bundles in the phloem occur under the skin, they support the conductive cells of the phloem and provide strength to the stalk. The glutted and bonded elementary lignocellulosic fibres are located throughout the whole length of the stalk parallel to the vertical axis. In these bundles, fibres are embedded in a pectic polysaccharidic network [17]

The elementary fibres are built from several layers: the first layer is the primary wall created during cell growth together with the secondary wall, which consists of three stratums and the thick middle layer and ensures the mechanical properties of the fibres, Figure 6. The middle layer is built from the long cellulose chain creating microfibrils. The size of the diameter of the microfibrils is approximately 10–30 nm. The microfibrils containing 30–100 cellulose molecules in the long chain give mechanical strength to the fibres.

The amorphous matrix phase in a cell wall is built of hemicellulose, lignin and sometimes pectin. The hemicellulose molecules are hydrogen-bonded to cellulose, they glue cellulose microfibrils creating the cellulose–hemicellulose network, which is the main structural component of the fibre cell. The hydrophobic lignin creates networks and bonds other molecular networks together resulting in an increase in the cellulose–hemicellulose composite stiffness, which reflects the fibre properties [18,19].

Cellulose, which is the most desired hemp fibre compound from a final application view of point, ensures the strength of the cell wall [20] and flexibility. Hemicellulose joins cellulose and lignin together [21]. Lignin enhances the strength and stiffness of the cell wall, lowers the sorption ability of the fibre and hinders chemical, physical and microbiological degradation [22]. Pectin occurs in the middle lamella between the cells of all types [23].

Hemp fibres have a structural and parietal organisation comparable to flax fibres in terms of biochemical composition, cellulose crystallinity or ultrastructural characteristics such as multi-layer wall organisation or the microfibrillar angle [25,26,27]. The hemp and flax elementary fibre structure are alike due to the similar agro-climatic conditions the plants grow in, both of the fibres are bast and the compounds in their chemical composition are the same. The similarity of the cylindrical structures of the elementary fibres of hemp and flax is presented schematically in Figure 6.

### 2.2. Hemp Fibre Chemical Composition

Hemp fibre, as lignocellulosic raw material, shows similarities to other bast fibres due to the comparable chemical composition of this fibre group. Fibres extracted from fibrous plant stalks contain cellulose, hemicellulose, lignin, pectin, waxes, fats and ash. Figure 7 illustrates the distribution of the main components of fibre within its structure.

The main macromolecular compound that occurs in bast fibres is cellulose in amounts ranging between approximately 40 and 80% of dry mass depending on the fibre type, variety and extraction method [30,31,32]. Cellulose is produced by synthesis of rosette terminal complexes (RTCs) at the plasma membrane in the plant. The protein structures of RTCs contain the cellulose synthase enzymes that synthesise the individual cellulose chains [33].

Cellulose is a polysaccharide and its high chemical reactivity is reached due to three hydroxyl groups –OH (Figure 8), which causes a high propensity for crosslinking during chemical modification.

Cellulose is hydrophilic, with a contact angle between 20 and 30 degrees and is insoluble in water and most organic solvents [35]. A feature of cellulose, the main component of fibre, plays a crucial role in the resulting properties of hemp fibre, determines fibre absorptivity, dyeability, ability for chemical modification as well as suitability for fibre processing.

Hemicellulose is one of a number of heteropolymers (matrix polysaccharides) (Figure 9) such as arabinoxylans, present along with cellulose in almost all terrestrial plant cell walls [36].

High crystalline cellulose is resistant to hydrolysis whereas hemicelluloses have random amorphous structures characterised by low strength. Hemicelluloses with cellulose bonded by the cross-linking of cellulose microfibrils, represent approximately 20% of the plant biomass [29]. Hemicellulose dominates the middle lamella of the plant cell and provides middle-ground support for the cellulose on the outer layers of the plant cell. Hemicellulose can also interact with lignin to provide structural tissue support of more vascular plants [36,37].

In the non-woody parts of terrestrial plants, pectin is a complex set of polysaccharides in the most primary cell walls. The main component of pectin is galacturonic acid, Figure 10. Pectin as an important element of the middle lamella that binds the cells together, and is also present in primary cell walls [38]. In bast fibres, pectin glues microfibrils and elementary fibres together, creating bundles, complexes called technical fibres. The presence of pectin in technical fibres is unwanted because it creates difficulties in fibre separation and further textile processing. The applied process of straw retting results in the reduction of pectin content in the hemp biomass and bundles of fibres.

Lignin is made up of a large group of organic polymers formed by cross-linking phenolic precursors [39]

Lignin occurs in the spaces between cellulose, hemicellulose and pectin components in the cell wall. The key role of lignin in cell walls is supporting the structure of the plant tissue: xylem tracheids, vessel elements and sclereid cells [40]. Lignin ensures the transport of water and aqueous nutrients throughout the plant stalk [41]. The polysaccharides present in the cell walls of the plant have hydrophilic characteristics, the crosslinking by hydrophobic lignin makes it possible for the plant’s vascular tissue to conduct water efficiently. Lignin occurring in the bast fibres causes the reduction of the fibre’s ability for water sorption. Lignin gives stiffness to the fibres, creating some difficulties during hemp processing.

The chemical composition of hemp fibres given by different authors differ substantially from each other. The percentage values of the share of the main chemical components in hemp fibres provided by different authors are shown in Table 1. The share of chemical components in the fibre depends on several aspects, e.g., hemp plant variety (Table 2), climatic conditions of growing, method of fibre extraction [30,42] and agricultural methods of plant cultivation including climatic conditions.

### 2.3. Hemp Fibre Properties

The parameters of hemp fibres, like linear density and length as well as fibre properties, are strongly related to the hemp variety, growing conditions and method of fibre extraction. This was proven by Vandepitte, based on the research conducted with the use of hemp varieties: USO 31, Dacia Secuieni, Bialobrzeskie and Futura 75 cultivated at the same time and conditions [47]. At the same, a study using seven varieties from different European origins (USO 31, Dacia Secuieni, Bialobrzeskie, Futura 75, Carmagnola Selezionata, Santhica 27 and Santhica 70) were explored. Despite significant variation between harvest years in straw yield, the quantity of the long fibres extracted held relatively constant. The tenacity of long hemp fibres was high overall and comparable to flax. Based on the study, it was reported that among seven tested hemp varieties, the Bialobrzeskie variety was characterised by the largest long fibre yield (Figure 11 and Figure 12) and delivered fibres with the best quality, e.g., lowest linear density and highest tenacity (Figure 13).

The differences in stem, bast fibre and primary fibre yield in relationship to type of cultivar was also investigated by Sankari [48] who conducted field experiments with 14 European cultivars of hemp (*C. sativa* L.) in Finland’s climatic conditions for three consecutive years. They proved the diversity of stem diameter, bast fibre content in the stem, fibre yield and mechanical fibre properties in relationship to hemp variety (Figure 14). Among all tested hemp varieties, monoecious Ukrainian cultivars USO 11, USO 31 and Polish cultivars Beniko and Bialobrzeskie showed very good features and were recommended for growing in Finland.

The diversity of the parameters of hemp fibres depends on many factors such as the type of variety, the growing climatic conditions and the applied method of retting, which all have effects on the chemical composition of the fibres, for example, the tensile strength and Young’s modulus of bast fibres increase with increasing cellulose content. This phenomenon is a reason that different authors of scientific articles give different values of hemp fibre dimensions (Table 3) as well as values of the mechanical properties of fibres (Table 4) [49].

The applied method of fibre extraction from the stalk determines the effectiveness of the removal of the gluing substances and dividing the technical fibres on smaller fibre complexes in pursuance of obtaining elementary fibres. The grade of the fibre separation is reflected in the fibre dimensions and properties.

Hemp fibres, similar to other natural fibres, have several bottlenecks resulting from their nature, the most significant is the lack of homogeneity. The lack of repeatability of fibre properties in batches delivered from farmers year by year is an essential disadvantage, the unevenness of fibre linear density, diameter and properties creates difficulties in the detailed design of processing the fibres and the planning of the quality of hemp products.

### 2.4. Moisture Sorption Ability and Fibre Swelling

Hemp fibres are characterised by a high ability for water and moisture sorption [51,52].

Different fibre separation processes deliver fibres with different surface characteristics, affecting the fibre surface morphology, which influences the specific fibre surface area. Research by [44] determined the ƺ-potential for the fibres and found differences in the degree of hydrophilicity in relationship to the applied process, whereby the green hemp fibres were more hydrophilic in comparison to retted fibres because green fibres contain more soluble components than retted fibres, which contain a relatively higher amount of waxes.

The bast fibre, as a lignocellulosic material, consists of mainly cellulose, hemicellulose and lignin in its chemical composition. The mechanism of cellulose or hemicellulose water absorption lies in the hydration process, which employs hydroxyl groups or –CH_2_OH groups of the host material [53]. The –CH_2_OH groups of the lignocellulose creates a relatively strong hydrogen bond and attracts other water molecules by weaker hydrogen bonding in the primary layer of the fibre. In the situation where the cell walls uptake water, the fibres swell until the forces of water sorption are counterbalanced by the cohesive forces of the cell walls [54,55]. A high content of crystalline cellulose in raw material negatively affected the accessibility and swelling of cellulose fibres [56]. In the case of flax fibres, the transverse swelling area is 47% and the swelling size of hemp fibres is very similar. The phenomenon of fibre swelling due to the high propensity for water sorption creates difficulties if hemp fibres are used as reinforcements of a composite. To reduce the negative effect of hydrophilic fibres on composite quality, specific modification processes for the fibres are applied [18,44,51,57,58,59].

Hemp fibres show a high ability for moisture sorption from surrounding areas, similar to other hydrophilic cellulosic materials [19]. The amount of bonded moisture by bast fibres depends on the surrounding air humidity. The values of hemp fibre moisture content in relationship to relative humidity of air is presented in Figure 15 [51].

Moisture hysteresis is a characteristic phenomenon of hemp fibres. Fibre moisture content depends not only on the relative humidity of ambient air but also depends on the direction of moisture changes. Values of a single measurement of fibre moisture made at the same humidity of ambient air but in the opposite direction form a loop of hysteresis, shown in Table 5. It is visible that when the fibre is moved from wet to dry conditions, desorption takes place, then the fibre moisture content is higher in comparison to the value obtained with the same air conditions where sorption appears, e.g., the fibre changes from dry to wet conditions.

Fibre moisture content has an effect on processing efficiency, and fibre linear density results from the swelling tendency of the fibres [60], shown in Table 6. The optimal value of moisture content for the hackling and scutching of the fibres suitable for further processing is between 10 and 12%.

## 3. Hemp Fibre Extraction from the Straw

The first process of the textile fibre value chain applied after hemp harvesting is the process making it possible to extract the fibre from the stalk. The approximate range of values of fibre content in the hemp stalk is presented in Table 7. Fibres in the form of phloem surrounding the stalk are located under the bark along the whole length of the stem. Their extraction requires employing processes which allow for loosening the stalk structure and reducing the occurrence of substances bonding the fibres with woody parts or together with each other. Loosening the stem can be conducted with the use of biological, physical, chemical or mechanical methods.

Hemp plants utilised for their fibre are harvested at the stage of fibre maturity or pre-maturity in the case of infantile hemp. The time of harvesting is determined approximately three–four months after sowing, the precise time is set for each case depending on the hemp growing climatic conditions; however, the time for hemp harvesting dedicated to fibre production takes place between the bloom and seed-set, e.g., between the time of harvesting for panicles collection in order to extract the Cannabidiol CBD and plant harvesting for the purpose of seed collection [8,32,50].

The multilateral approach to hemp fibre extraction presented in Figure 16A–D shows the possibilities of applying different tracks of the value chain. Retting, the first process of fibre extraction, determines fibre quality. The main idea of the process is the biological degradation of pectin and other cementing compounds such as hemicellulose and lignin that bind the bast fibres and fibre bundles to other stalk tissues and, thereby, separate fibres from non-cellulose materials [61,62,63,64]. A historically well-known method of fibre extraction from the hemp straw was the water retting process [65], which, due to high toxicity to the environment has been replaced by dew retting. However, during the last few decades, the new water retting methods have been developed with respect for environmental protection [66,67]. The relatively younger retting methods are enzymatic retting [68] or osmotic degumming [69]. Regardless of the applied method, after the retting, it is necessary to conduct mechanical processes such as breaking and scutching to separate the fibres from the woody parts and divide the technical fibres into smaller fibre complexes (Figure 16A). Scutched long fibres used for high quality textile production are further processed on a hackling machine. Short fibres created during scutching have to be carded to prepare thicker yarns for thick fabrics, for example, upholstery, and other woven applications, as well as raw materials for nonwovens and technical products.

To avoid the retting process, the stalks collected from the field can be decorticated [70]. This mechanical process aims to break the stem, which allows for fibre separation. However, decortication delivers low quality, inefficiently divided bundles of fibres, with a high impurity content, making the use of this raw material unsuitable for textile purposes (Figure 16C). Pure decorticated fibres are used for technical purposes, such as mats, nonwoven and other. To obtain clear fibres with low linear density, it is necessary to conduct the degumming process following decortication to remove the pectin-bonded fibres. This value chain is presented in Figure 16B [71]. Degummed decorticated hemp fibres are processed on a carding machine in order to produce cotton-like or wool-like hemp fibres in terms of their dimensions and final application. There is the possibility to reverse the processes: decortication → degumming → carding (Figure 16B) and apply retting → decortication → carding (Figure 16D). In both cases, produced fibres are carded and shortened to make them suitable for spinning yarn with the use of a cotton or woollen spinning system.

Methods of hemp fibre extraction correspond to methods developed for flax fibre, including retting as a key process of fibre extraction or the mechanical extraction with the use of decortication. The aim of retting, regardless of the applied method, is the degradation of the parenchyma cells, as well as in the middle lamella between the fibres [64,65] to remove the non-cellulosic components and to separate the cellulose fibres.

Among the retting methods applicable for hemp, it is necessary to list five methods of significant importance, which are currently used and which have the most potential for further development and improvement in terms of a sustainable approach:Dew retting;Enzymatic retting/fungal retting;Chemical retting;Water retting;Physical retting.

The type of process applied for hemp fibre extraction has an effect on the chemical composition of the fibres and the resulting properties of the fibres [30]. Table 8 shows the differences between the values of content of the individual compounds in the hemp fibres of the Bialobrzeskie variety extracted from the straw with the use of different methods: dew retting, decortication, osmotic degumming and water retting. The water retting method is the most effective in removing the non-cellulose substances from the hemp biomass. The water-retted hemp fibres, due to the highest cellulose content, are the most suitable for textile purposes among all fibres achieved with use of other tested extraction methods.

### 3.1. Dew Retting

Currentlty, dew retting is the most common method used for flax and hemp. There are three methods of dew retting:-On the field in ambient environmental conditions;-Retting in the greenhouses under controlled ambient conditions;-Fost retting.

#### 3.1.1. Retting on the Field in Ambient Environmental Conditions

Dew retting on the field is the cheapest method because it does not need the use of special equipment apart from overturning and harvesting machines. The process is characterised by low energy and no water consumption, only sun energy, rain and ambient air with diversified humidity are needed, although straw laid on the field occupies the soil surface for several weeks making the use of the field unavailable for other purposes. From the agricultural perspective, field retting has a positive effect on soil fertility because of degradable natural residues that appear during the process [62,72,73,74].

The biggest disadvantage of dew retting is the impossibility to control the process parameters such as temperature and air humidity, which have an effect on the process duration as well as fibre yield and consequently, on fibre quality [63]. Uncontrolled weather conditions determine retting parameters, making it difficult to predict and repeat the desired quality of retted fibre features in consecutive years.

Dew retting is a biological process, where bacteria and fungi are employed to degrade non-cellulosic components of the fibre. The study cond ucted by Ribeiro [61] proved a higher bacterial diversity in comparison to fungi during the hemp dew retting process; however, proportions of occurring microbial species depend on the type of ground, harvest dates and retting durations on fields, and the differences are statistically significant. The bacteria species: *Escherichia coli*, *Pantoea agglomerans*, *Pseudomonas rhizosphaerae*, *Rhodobacter* sp., *Pseudomonas fulva*, *Rhizobium huautlense* and *Massilia timonae* as well as fungal sequences related to the genera *Cladosporium* and *Cryptococcus* occur in the majority during dew retting [69]. *Pseudomonas* species are particularly important in the decomposition of pectin in plant fibres in both aerobic and anaerobic conditions [75,76]. Fernando [77] broadened the Akin [62] study and found that fungi, considered so far as the main player in dew retting, achieves their highest activity in the early stage of retting whereas bacteria outcompeted the fungi in the longer retting period. Fungal species are able to destroy the cuticular layer with extracellular cutinises by hyphal entry through damaged areas [73]. A long-lasting retting process results in the employment of a larger bacteria community and bacterial mycophagy, including extracellular necrotrophic and endocellular biotrophic behaviours, that influences bacterial activity. The activity of fungi causes the degradation of pectin-rich cells, e.g., parenchyma and, consequently, easy separation and defibration of the bast fibres from the complex stem cellular structure after two weeks of retting. Bacteria, together with fungal species, metabolise parenchyma cells between fibre bundles with pectinolytic enzymes and hemicellulase.

The duration of dew retting means the duration of microorganisms acting on fibres has an effect on the fineness of scutched hemp tow: the longer time of retting ensures lower linear density of the fibre (Figure 17). After six weeks of retting, a reduction in the linear density of the fibre in comparison to non-retted fibre is nearly 50%. The type of soil had no statistically significant effect on fibre fineness [61].

Réquilé [27] confirmed that a hemp retting effect that ensures the required quality of fibres can be achieved by controlling the duration of the retting process conducted in different weather conditions, e.g., the intensity of solar radiation, rainfall and ambient temperatures. In the case of rainfall deficiency or low air temperature, the period of retting should last longer to make it possible to reach a suitable level of retted biomass. The conditions of the process and retting level of the stem have to be carefully monitored because the dew-retting process has an impact on the tensile properties of elementary hemp fibres (by degrading crystalline cellulose I).

Many microorganisms responsible for the decomposition of parts of the stem accompanying the fibre could have a special role in the retting process. It means that the additional use of bacteria and fungi selected from the perspective of adjustment to the type of soil and weather conditions can shorten the duration of the and improve fibre quality.

#### 3.1.2. Retting in the Greenhouse under Controlled Ambient Conditions

Conditions of the field retting process are difficult to predict and impossible to design precisely, which results in unpredictable fibre yields and features. To avoid dependence on the effectiveness of retting with regards to weather conditions and controlling the role of microorganisms in the decomposition of hemp stems and non-cellulose fibre components, the retting in greenhouses under controlled ambient conditions are conducted [78,79] The method of greenhouse-retting, even its experimental performance is adequate for hemp retting targeted for special use and allows for significant changes occurring during the dew retting process, especially in the case of the stem surface colour resulting from the microbial colonisation causing the selective degradation of the cell walls of the bast tissues parenchyma and the decohesion of the bast tissues. Law [79] found that hemp harbour a resilient cohort of microorganisms that can be present at the harvest stage and continue to persist throughout the retting process.

Greenhouse dew retting can only be used for limited hemp cultivation addressing special targets, which require controlled process conditions ensuring precisely designed fibre properties determined by the final application. Apart from this, experimental retting conducted with the use of a greenhouse can be a source of new knowledge about dew retting mechanisms and selective micro=organisms’ role in hemp biomass decomposition.

#### 3.1.3. Frost Retting

In Nordic countries, due to specific weather conditions, e.g., low temperatures and high humidity during autumn, typical hemp dew retting is not effective. The drying of wet stems laid on the field in ambient conditions is a risk due to the possibility of mould formation and for this reason the application of an additional drying process is necessary even it is associated with high energy consumption [80,81]. Hemp harvesting in spring and frost-retting in these countries has been developed, where the fibre separation from the stem is induced by the daily changes in temperature above and below zero during the springtime. The water freezes and melts in the plant cell structures, which leads to an enlarging movement that loosens the bast fibre from the stem. The quality of frost-retted fibres is not too high but is enough for technical application.

### 3.2. Water Retting

The oldest method of bast fibre retting is water retting practised in Western and Eastern Europe, Asia, in Egypt and other countries located in different continents, where the climatic conditions allow for fibrous plant cultivation. According to this method, the bast fibre stems were pulled and submerged in water tanks [65], pools or natural water reservoirs such as lakes or rivers for five to seven days where anaerobic bacteria caused the degradation of primarily pectin. In the artificial water tanks constructed for retting, the process parameters such as water temperature and selected microorganisms used to improve the process could be controlled. Tanks were aerated to modify the bacterial consortium and thereby the bacterial metabolism, reducing the problems of pollution and stench from anaerobic metabolism [62]. Available air initiates the retting through the development of aerobic bacteria from the *Bacillus* or *Paenibacillus* genus and then the process is continued through anaerobic bacteria (from the *Clostridium* genus) [82,83]. Pectinolytic microorganisms developed during water retting are *Bacillus* spp., dominant from 10 to 40 h after the start of the process, and are succeeded by spore-forming anaerobic *Clostridium* spp. when oxygen concentration in water tanks becomes lower [83,84,85]. The shortest water retting time (up to 4 days) for hemp retting is possible when in the tanks both aerobic and anaerobic bacteria are inoculated simultaneously [85]

In the past, the stems after retting were exposed to the rays of the sun to dry and bleach on the field [63]. Sun drying is a long process dependent on weather conditions and related to land occupation. However, the drying of water-retted stems in dryers is a process characterised by high energy consumption and the need for additional logistic solutions, which have a negative environmental and economic impacts [75,86,87,88,89].

The long fibres extracted from the stem with the use of water retting are characterised by high quality in terms of fineness, mechanical properties and spinnability, which make them excellent raw materials for textiles, including apparel applications. Traditional water retting was not practiced between the 1960 and 1990s of the last century because it was prohibited by most countries due to the generation of large amounts of wastewater and high freshwater consumption, the high pollution of soil and air as well as the stench residing in water-retted fibres. However, water retting is still conducted in some countries of Asia and in Egypt mainly on Nile Valley and Delta soils [66,90,91].

The duration of the process required for adequate retting under water (1–2 weeks) is shorter than field retting, and the dependence on weather and geographical location is minimized. Moreover, process parameters such as temperature and pH levels can be maintained to reach optimal levels in artificial water bodies [66,67]. The water retting process was the most recommended retting process for high-quality bast fibre production [62].

Apart from freshwater retting, it is possible to conduct hemp water retting in the sea. Zhang [92] confirmed the possibility of retting hemp in the sea in China. Sea water retting of hemp caused a significant reduction in pectin and hemicellulose content compared with the raw hemp fibre resulting from the removal of non-cellulosic gummy materials.

### 3.3. Enzymatic Retting

The enzymatic retting of hemp fibres has attracted the attention of researchers and stakeholders due to the significant shortening of the process duration, the possibility to control retting conditions and manage its efficiency to obtain fibres with specific properties [43]. Enzymes are proteins that act as biological catalysts which accelerate chemical reactions, e.g., enzymes increase the reaction rate by lowering its activation energy. Enzymes hydrolyse pectin and gum material in the stem causing their degradation within a short time, e.g., between 12 and 24 h, ensuring efficient fibre separation and division into small complexes. Specific fibre properties can be achieved for different applications by varying the retting duration and type of enzymes used. The disadvantages of enzymatic retting are the risk of a decrease in fibre strength and the high cost of enzymes.

Enzymatic retting of flax stems is a well-known method described in many scientific articles [93,94,95,96,97], although studies dedicated specifically to enzymatic hemp retting are limited. Enzymatic hemp retting, in spite of high enzyme costs, is developed by researchers and stakeholders due to the positive impact of this method on achieving hemp fibre quality suitable for composite reinforcement. Microbial retting conducted with the use of the white rot fungi *Ceriporiopsis subvermispora* and *Phlebia radiata* Cel 26 causes high selectivity for pectin and lignin degradation keeping the cellulose content on the high level in the retted hemp fibres [68]. The pectin and lignin mainly located in the outer part of the fibres are assumed to be extracted and degraded by pectinase and peroxidase enzymes produced by the fungi.

A study on the suitability of hemp fibres obtained with the use of three different methods, classical field retting, controlled fungal *Phlebia radiata* Cel 26 retting and pure pectinase treatment for the production of fibre/epoxy composites, were conducted by Liu et al [98]. They found that composites reinforced with fibres obtained by enzymatic retting with the use of pure pectinase showed the highest composite strength among all the tested hemp fibres. However, fungal retting could degrade non-cellulosic components from hemp fibres at the highest selectivity. Both methods, fungal and enzymatic retting, deliver fibres with lower porosity content in comparison to dew retting, and in the case of the use of the fibres for composite reinforcement, result in a more homogeneous composite and better fibre–matrix adhesion.

In terms of the sustainability approach and LCA analysis, the enzymatic retting using endo-polygalacturonase and pectin-lyase requiring pre-treatment by a hydrothermal process to facilitate enzyme penetration and to reduce enzyme consumption, showed higher environmental impacts in comparison to traditional field retting due to increased energy and material consumption, which are not counterbalanced by reductions in other life cycle stages [99]. From the other side, the enzymatic treatment targets more effective non-cellulosic components removal, delivers hemp fibres with good mechanical properties and lower porosity, which, when applied to reinforce composite, helps to reduce the overall environmental impact of the hemp fibre composite.

### 3.4. Chemical Treatments

Chemical processes used for hemp retting ensures a significant shortening of the retting period and lower costs in comparison to enzymatic processes but often yield more coarse fibres [73]. Chemical retting is commonly used for flax stems with complex agents and detergents buffered to a high pH with alkali. For chemical treatment addressing retting, the following agents are the most often used: EDTA, diethylenetriaminepentaacetic acid, oxalic acid, tetrasodium pyrophosphate and sodium tripolyphosphate; the alkalis commonly employed are NaOH, KOH or Na_2_CO_3_; sodium dodecyl sulphate is widely used as a detergent [100,101].

The effectiveness of chemical retting at a high pH depends on the chelator type and level, sodium hydroxide levels, as well as the plant condition and maturity; however, when the factors are optimised, chemical stem treatment can deliver fine fibre yields.

Currently, there is an urgent need to protect the natural environment, forcing pursuance to avoid or reduce chemical use, causing lower interest in bast fibre chemical retting. The pro-green approach requires looking for processes characterised by low energy consumption and no chemical use [102].

### 3.5. Physical Retting

#### 3.5.1. Steam Explosion

Steam explosion (SE) is a process in which the lignocellulosic biomass is treated with hot steam (180 to 240 °C) under pressure (1 to 3.5 MPa) and then proceeds from decompression to atmospheric pressure. This process causes a breaking off of the fibre’s rigid structure, changing the biomass into a fibrous dispersed solid [103,104,105]. Hydrolysis of the glycosidic bond and hydrogen bonds between the glucose chains results from the sudden release of pressure which generates shear force.

The steam explosion process is often supported by additional chemical treatments. Hemp fibre extraction conducted with use of SE combined with alkali hydrolysis is able to separate fibres in an efficient way and delivers approximately 92% of elementary fibres from the whole fibre biomass [106]. Processing using an alkali treatment and steam explosion technique can be also preceded by the preparation of fibre, e.g., cutting into uniform size of approximately 2 mm lengths, which allows steam explosion and chemicals to penetrate deeper into the inner layers of the fibres [107]. SE combined with chemical modifications of hemp fibres effectively removes impurities, pectin, hemicelluloses, lignin and waxes resulting in the decrease of the fibre diameter. The effectiveness of the processes depend on the treatment conditions, parameters and composition.

#### 3.5.2. Osmotic Degumming

Osmotic degumming of bast fibres is a method used for fibre extraction based on natural physical phenomenon such as water diffusion, osmosis and osmotic pressure. During the water flow through the stem mass, water diffuses into the stem causing it to swell, resulting in increasing hydrostatic pressure within the stem and the epidermis in the longitudinal view is broken; however, there is no shortening the fibres. The pectin becomes diluted and dissolved along with other substances present in the phloem in the flowing water [69]. Osmotic degumming used for bast fibre extraction can deliver well-cleaned, high quality, soft, long fibres with bright colour without odour typical for water-retted fibres. The process is supported by the application of ultrasound in a water bath, which significantly improves the effectiveness of fibre degumming. To make the method more sustainable, the process is equipped with a closed water system; however, the use of energy from renewable resources should be considered in the future.

## 4. Hemp Value Chain from Fibre to Yarn

Increasing interest in hemp textiles is due to an appreciation for the advantages the hemp plant can provide and growing societal awareness in terms of creating demand for bio-based products and protecting ecosystems to ensure sustainable solutions for a circular bio-economy [108]. This approach provides an opportunity for businesses and stakeholders to develop the hemp fibre sector by rebuilding and creating new companies which are able to produce hemp textiles. The bottle neck for the hemp textile value chain is the spinning system. The traditional linen spinning technology shown in Figure 18A seems to be the most suitable for hemp yarn production for textile/clothing purposes [31,70,109] but the system requires a very specialised machine park containing machines which have not been produced for several decades and are difficult to find on the market [11]. Other existing spinning systems, such as cotton or wool spinning technologies, can be adapted for hemp fibres, making the production of pure hemp yarn or blended hemp with relevant natural or man-made fibres possible. The systems are graphically presented in Figure 18B–D.

The linen spinning system, shown in Figure 18, consists of several highly specialised machines dedicated to the processing of retted bast fibres, while the hackling machine for long fibres and the wet spinning frame are markedly different from relevant machines used in other spinning systems [31,70,110]. The linen spinning system [65] needs hemp fibres prepared according to a determined procedure: the harvested straw must be retted with the use of one of the possible retting methods, then the dry stem has to be cut to 0.8–1 m lengths and scutched [111]. The scutching output are long fibres in the shape of tresses ready for hackling and a mass of hemp tow [112,113]. The clean, hackled long fibres are drawn and mixed to be laid parallel in formed slivers suitable to make roving. Hemp yarn is spun from bleached or raw-grey roving. From 1 ton of raw hemp straw, it is possible to produce approximately 70–80 kg of long hemp fibre thin yarn characterised by the best quality in comparison to other spinning technologies, used for clothing and advance textiles. Scutched tow or hackled tow are used for the production of yarn with higher linear density, while hackled tow can be processed with the use of wet or dry spinning and, alternatively, they can be cottonised to spin on the rotor spinning system [114]. Although the linen spinning system should be treated as a basic well-known technology ensuring the best quality of yarn from hemp long fibres, many challenges are associated with the system. The first challenge causing serious limitation in the use of the system is difficulty to complete the technological line due to the lack of producers interested in the construction of the machines. The hackling machine is unique, it can be used only for flax/hemp long fibre processing, it is not possible to include it in other fibre spinning systems. The limited demand for the machines is determined by the niche character of the flax/hemp sector. The share of flax/hemp fibres in the world production of all types of fibre was less than 1% in 2019 [115]. It means that the development and production of new hackling machines for very niche hemp fibres is economically unprofitable.

The traditional technology of production of linen/hemp yarn, presented in Figure 19, is dedicated to retted fibres [71]. To avoid a costly, long lasting, labour-intensive and non-inert environment for the retting process, the decortication system was introduced several decades ago. The decortication process covers the mechanical extraction of fibres from raw non-retted stalks by breaking the woody parts and mechanically separating the fibres. Decortication delivers one type of fibres in terms of length with a high impurity content. Obtained fibres are in the shape of big complexes of glutted together fibres fixed with pieces of broken stem. The use of the simple decortication process for the extraction of bast fibres from the straw is well-known and commonly applied for technical purposes where the good quality of fibres and low impurity content are not required. There are two ways to improve decorticated fibre quality. First, the application of the retting process before decortication, when the main glutted substances such as pectin are destructed by bacteria/fungi/enzyme activity to ensure more effective mechanical fibre separation and division into small fibre complexes (Figure 18B). The second way to obtain good quality decorticated fibres is the application of degumming after the decortication process (Figure 18D) [116]. Degumming of the decorticated fibres can be conducted with the use of different methods, for example, by a biotechnological process where fibres are washed with 0.5% soda solution and treated with the pectinolytic strain *Geobacillus thermoglucosidasius* PB94A and then dried [117]. Another degumming method, with confirmed efficiency in dividing fibres into almost elementary ones, is the hydrodynamical process conditioned by a flow of water at 30 °C through the fibres formed into a reeled sliver, and then by moving in open degumming devices with water supported by ultrasound [116].

Whichever of the two combined methods of retting → decortication or decortication → degumming is used, it is necessary to conduct further mechanical processes, including carding, to tailor fibre the dimensions of needs determined by the final application, for example, cottonisation or woollenisation (Figure 18).

The use of the cotton spinning system for cottonised hemp fibres makes it possible to produce pure hemp yarn or hemp blends with other cotton-like fibres by using a ring cotton spinning frame or rotor spinning. The wool-like hemp fibres can be mixed with wool and processed on the woollen spinning system.

The above-described diversity of methods of hemp yarn production makes it possible for stakeholders to research the spinning technology available and adapt it for hemp fibre according to their infrastructural potential and determined target, although it is necessary to be aware that each technology delivers hemp yarn characterised by different quality parameters.

Consideration of economic aspects of hemp production should cover a holistic approach, including plant cultivation with agrotechnological costs such as fertilisation, watering, land occupation, yield of plants, fibre and by-products (shives), costs and productivity of fibre extraction processes as well as further processes leading to yarn production. The price of hemp fibre also depends on climatic/weather conditions, which have effects on fibre yield and is usually diversified in subsequent years and in countries where the hemp is growing. The report covering the comparison of the economic aspects of hemp and cotton fibre production in the USA in 2018 provides data of the total agricultural cost for the fibre production with regards to fibre yield [118]. The cost of hemp fibre production in the case of medium yields of fibres was in the range between 1328.92 and 357.25 USD/t, while cotton cost production for medium fibre yields was between 3556.48 and 2845.15 USD/t. The value of the total cost calculated for the medium value of final hemp fibre yields was approximately 77.63% lower in comparison to the respective cotton fibre. In the case of the low value of the total cost from the range mentioned above assessed for the high final fibre yield, the value of hemp fibre comprises 1/12th of the cost per metric ton of cotton fibre. It can be concluded that the production of hemp fibres can be more profitable than cotton in the case of medium and high fibre yields.

The economic aspect of hemp fibre production in Turkey, discussed by Ceyhan [119], showed that farm-level hemp production and processing with the use of modern technology was economically viable, while the viability of processing hemp by conventional technology was not profitable in Turkey due to low productivity and the small scale of the production.

The literature records discussing economic aspects of the whole fibre value chain are very limited and there is no information on the economic viability of industrial hemp targeted for textiles.

Comparing the profitability of hemp yarn production is difficult due to the diversity of methods possible to use for the same value chain, e.g., there are many retting technologies, many available combinations of technological lines and different machines providing varying levels of productivity. The traditional spinning systems are characterised by low productivity, nevertheless modern technological solutions are still on a laboratory scale or semi-commercial scale. The cost of production depends on the type of production system applied and should be evaluated individually for each case. The lack of production of specialised industrial machines dedicated to hemp fibres results in the high diversity of technological lines used and makes it impossible to evaluate the universal economic aspects of hemp yarn manufacturing [120].

## 5. Challenges and Opportunities

World production of all textile fibres reached 110 million tonnes in 2019. Natural fibres comprised 30% of the total fibres produced in the world (Figure 20) [114,115]. In 2019, oil-based fibres such as polyester, acrylic and nylon accounted for 69 million tonnes of production, including 50 million tons of filament and 20 million tons of staple fibres.

In 2019, the production of hemp fibre and tow was 60,000 tons, the percentage share of hemp with reference to all natural fibres was 1.7%, but hemp was only 0.05% with respect to all world textile fibre production.

The fact that hemp is a niche fibre means that the development of the hemp fibre sector meets many challenges, with the most significant weakness being the lack of specialised modern machines required to complete the technological line. Hemp spinning is not an attractive direction for producers of machines, such as the hackling frame dedicated to the long fibre process, which guarantees obtaining the best quality hemp yarn and final product. Existing traditional equipment usually used for yarn production is characterised by low productivity, which makes the process economically unviable.

The low profitability of hemp textile production, which is conditioned by both agricultural and industrial activity is a challenge for stakeholders. The high yield of hemp fires and high productivity of processes allows for improving economic results in the hemp industry. Developing biotechnological and agrotechnological methods, which make it possible to cultivate and harvest plants delivering high yields of fibres with specified properties, the transfer of knowledge from research centres to industry in terms of fibre processing, the building of a hemp machine-rich market allowing for the creation of innovative technological lines as well as the creation of new business models are necessary to make hemp textile fibres economically competitive in comparison to other natural fibres. The strengths for a future-proof business model are circularity and zero waste in the hemp industry, which can help mitigate climate change.

Increasing profitability of hemp textile production allows for facing social challenges. Population ageing is a challenge for the European economy and society. This phenomenon is also visible in the hemp sector, where a generation gap is observed. Employees in the European bast fibre sector are experienced workers aged 50+, as the interest of young people to work in hemp sector is very low. It is necessary to create an education system to ensure competent farmers and staff to operate tools or machines, implement new technologies and develop the hemp industry.

The education system should also aim to increase social awareness in terms of the sustainability of bio-products and the effects of hemp on the environment and human life.

Sustainability of hemp fibre is the main strength of the whole sector. The huge amount of non-degradable synthetic oil-based fibres such as polyester, acrylic and nylon produced every year constitutes hazards for the environment, even with some of them being able to be recycled, redistributed, refurbished, remanufactured, repaired or reconstructed for use again. Nevertheless, synthetic fibres eventually become non-degradable waste. The launched European Green Deal Strategy targets the protection of our globe and human environment against climatic changes and global warming, with the aim of reaching climatic neutrality in 2050 [5].

The pursuit of plastics elimination in many sectors, including everyday life, and replacing non-renewable raw materials, e.g., construction, plastics, textiles, with natural ones will contribute to the reduction of greenhouse gasses emissions and, consequently, decarbonization [108]. This fact creates a serious opportunity for the multi-perspective development of hemp, which is a model plant in terms of suitability and bioeconomy. The possibility to cascade the use of the plant delivering natural raw materials for production of bio-products in different sectors of the economy is attracting the increasing interest in hemp growing and processing. Hemp is the most suitable plant to cultivate in this time of need concerning the reduction of CO_2_ emissions [6,7]. Hemp cultivation protects agricultural areas against the loss of biodiversity, improves the productivity of the soil, removes heavy metals and can be used for soil remediation and reclamation in industrial areas. Additionally, 1 ha of hemp plantation absorbs approximately 10t of CO_2_ from the atmosphere every year.

The multi-perspective approach to hemp plants contributes to the development of the hemp industry, including the hemp textile sector. Apart from the sustainability of hemp growing, processing and bio-product manufacturing, it should be emphasised that hemp apparel and functional textiles create the optimal environment for wearers, ensuring physiological comfort in everyday life and positive effects on the skin thanks to antioxidant and antibacterial activity. Hemp fibres could produce textiles with barrier activity against UV radiation due to their unique chemical composition containing lignin and phenolic acids [30,121]. There is significant demand for hemp textiles due to specific hemp properties, which has been observed worldwide on the textile markets.

Optimization of the environmental impacts associated with the whole hemp value chain from cultivation to the technological processes and up to the production of hemp textiles, focusing on the quality of the end products is a challenge for the hemp sector [71,111]. Assessment of the life cycle, including all elements such as land occupation, pesticide use, direct water use, non-renewable energy and chemical use, gas emissions, processing productivity and the evaluation of the quality of the by-products at each stage of the chain should be used by stakeholders to select the most suitable and profitable technological chain corresponding to the infrastructural potential of the company and to ensure the sustainability of the processing.

## 6. Conclusions

Significant demand for hemp textiles observed on the world market and the multi-perspective positive environmental impact of hemp cultivation, processes and use provides prudent and opportunistic conditions to develop the hemp sector. The hemp value chain presents a model bioeconomy solution meeting the assumptions of the European Green Deal.

The diversity of possible methods for hemp fibre extraction that conditions the fibre chemical composition and properties gives stakeholders the opportunity to select technology that would be the most suitable for their capability and final application.

Based on the reviewed methods of fibre extraction, it can be concluded that three methods with the lowest environmental impact will be developed in the future, e.g., dew retting, water retting with a circular water system and renewable energy use as well as mechanical extraction with the use of decortication. The first two methods ensure good quality of long fibres, the last one provides raw materials suitable for cottonisation or technical application. The use of enzymatic retting is limited due to the high price of enzymes, while chemical retting is not neutral to the environment.

The most feasible direction for the development of hemp yarn production is the adaptation of the cotton spinning system to cottonised hemp fibre features. This is conditioned by commonly available machines, which are usually used for cotton and cotton-like fibre processes. The quality of yarn made of cottonised hemp fibres is lower in comparison to yarn spun from long hemp fibres in terms of yarn uniformity and its mechanical properties which result from huge differences in fibre length. The cottonisation of hemp and the applied cotton spinning system causes hemp to lose its inherent properties like high tensile strength, cool touch, resistance to pilling and bioactivity. Alternatively, textiles made of cottonised hemp can be more resistant to wrinkling. The productivity of the cotton spinning system is much higher than productivity of the traditional linen spinning system but only on the condition that the cottonisation of hemp fibres is conducted with high accuracy. In this case, there is a chance to develop the production of hemp textiles for relatively low costs in the near future. The production of the best quality hemp yarn from long fibres will be dedicated to exclusive premium clothing with high prices due to the high cost of production.

Significant factors provide proof of hemp sustainability and strengthen the chance for the development of the hemp textile sector as a part of holistic hemp business. These include the multi-perspective environmental benefits concerning hemp cultivation, such as the absorption of CO_2_ from the atmosphere, improvement of soil quality and enhancing biodiversity, helping mitigate the effects of climate change and restore healthy ecosystems, as well as the possibility of using every part of the hemp plant for different purposes and wasteless industrial hemp processes, where each by-product is a valuable raw material for many sectors of the bioeconomy.

## Figures and Tables

**Figure 1 materials-15-01901-f001:**
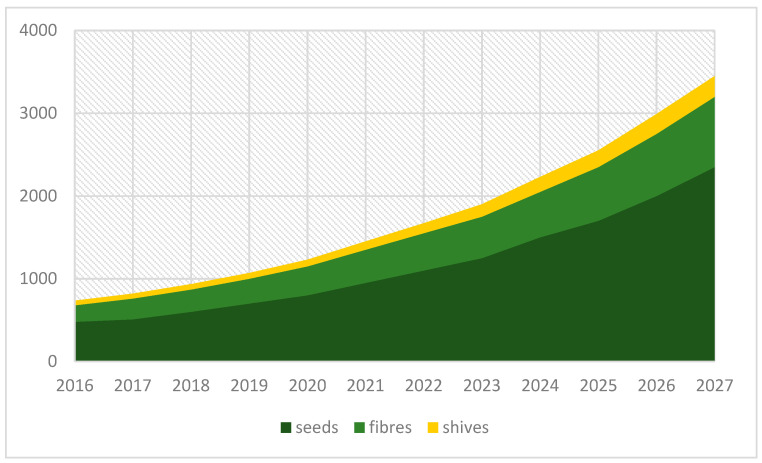
Forecast of industrial hemp market size in USA, by product, 2016–2027 (USD mln), based on [10].

**Figure 2 materials-15-01901-f002:**
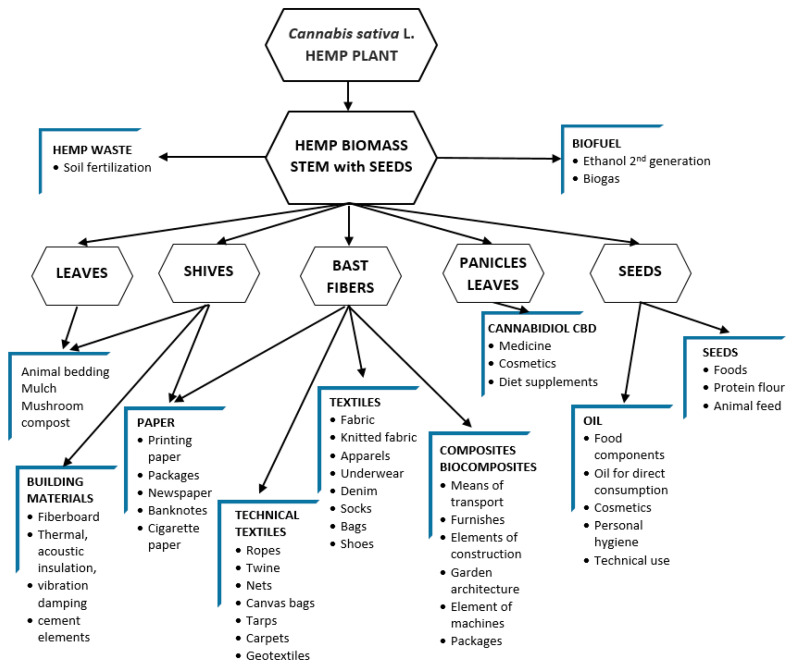
Schematic performance of multidirectional potential of the use hemp raw materials, based on [12].

**Figure 3 materials-15-01901-f003:**
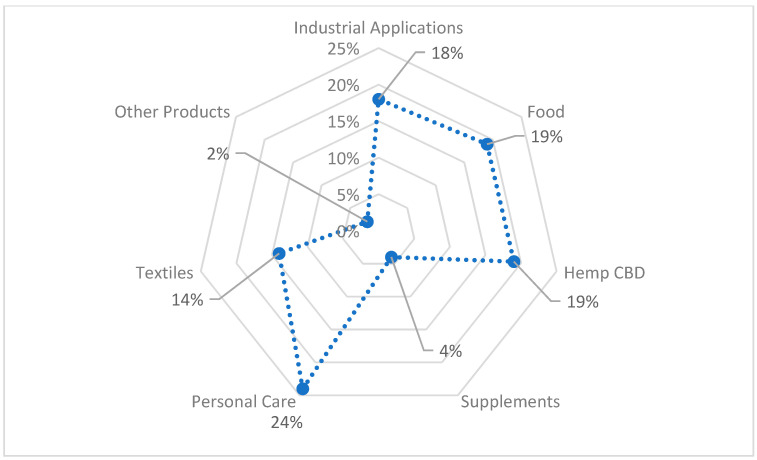
Hemp-based product sales by category, 2015, based on [13].

**Figure 4 materials-15-01901-f004:**
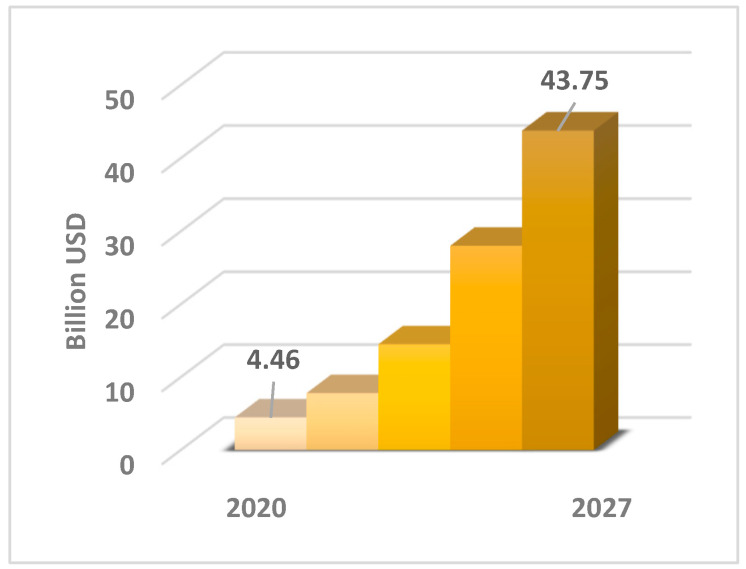
Forecasted size of global hemp fibre market up to 2027 based on [15].

**Figure 5 materials-15-01901-f005:**
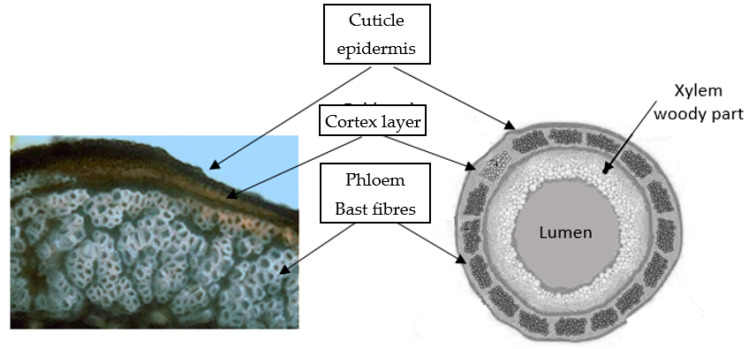
Cross-section of hemp stalk, based on [24].

**Figure 6 materials-15-01901-f006:**
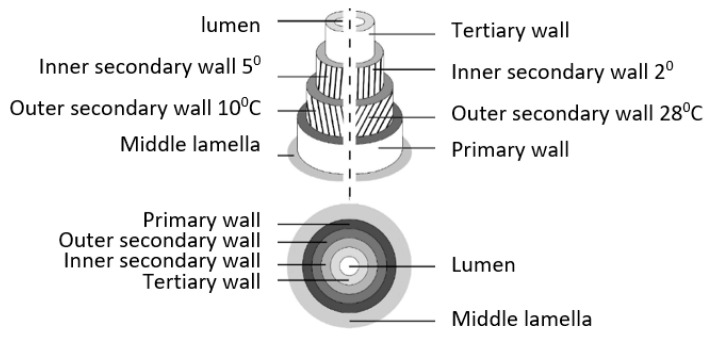
Structure of the elementary hemp fibre. (top image) Longitudinal section; (bottom image) Cross-section of fibre [28].

**Figure 7 materials-15-01901-f007:**
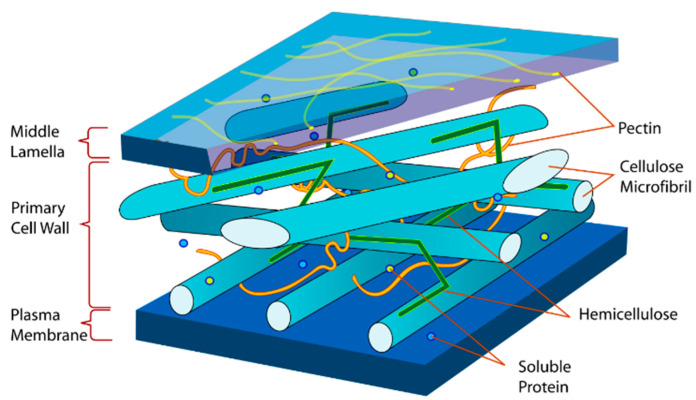
Schematic image of the section of a hemp cell wall; hemicellulose in green [29].

**Figure 8 materials-15-01901-f008:**
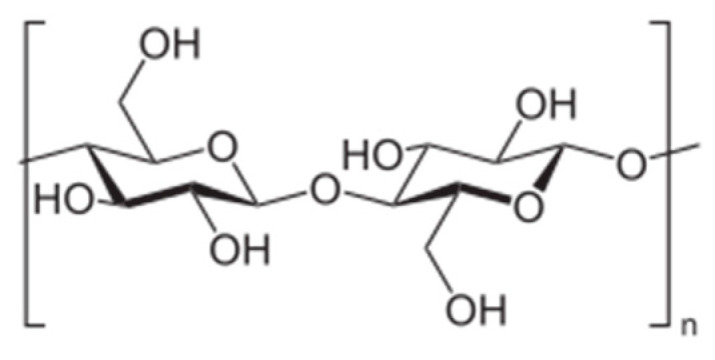
Structural formula of cellulose [34].

**Figure 9 materials-15-01901-f009:**
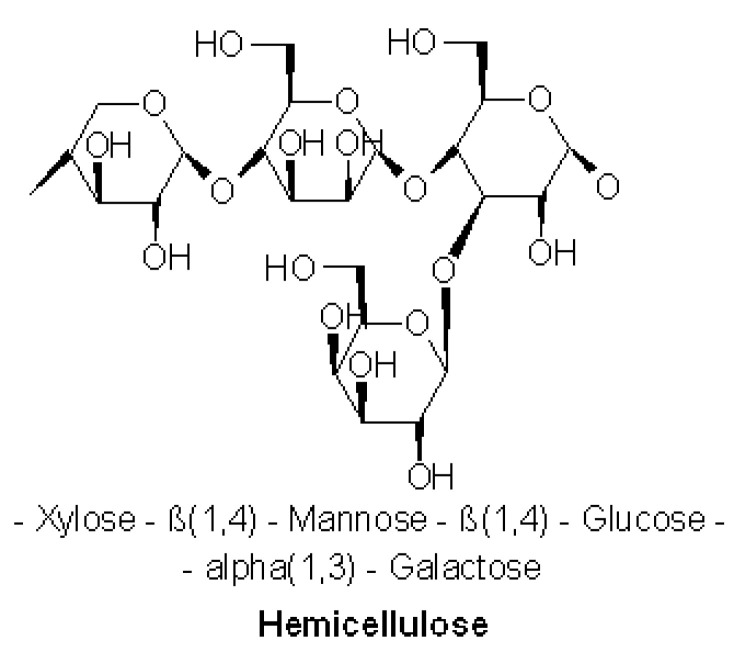
Most common molecular motif of hemicellulose [34].

**Figure 10 materials-15-01901-f010:**
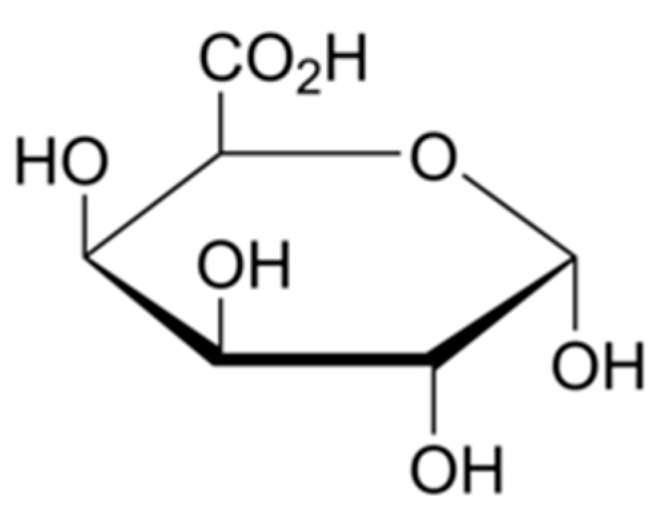
Structural formula of galacturonic acid, the main component of pectin [34].

**Figure 11 materials-15-01901-f011:**
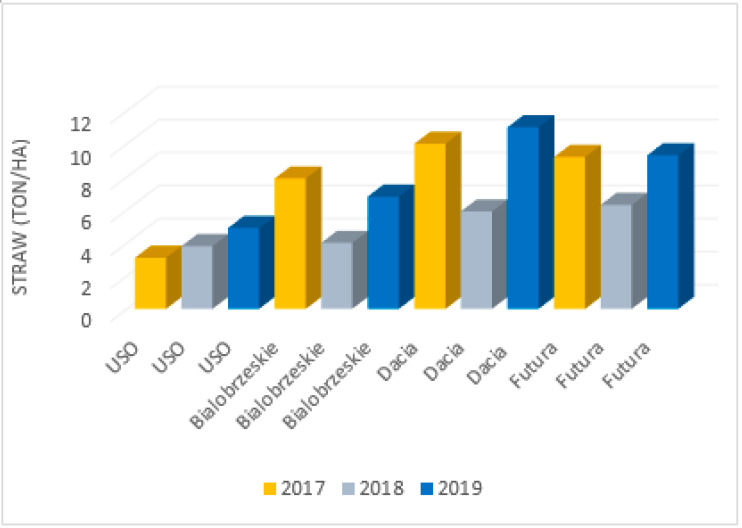
Mean straw yield determined for different type of hemp cultivars: USO 31, Dacia Secuieni, Bialobrzeskie, Futura 75, cultivated for three consecutive years, based on [47].

**Figure 12 materials-15-01901-f012:**
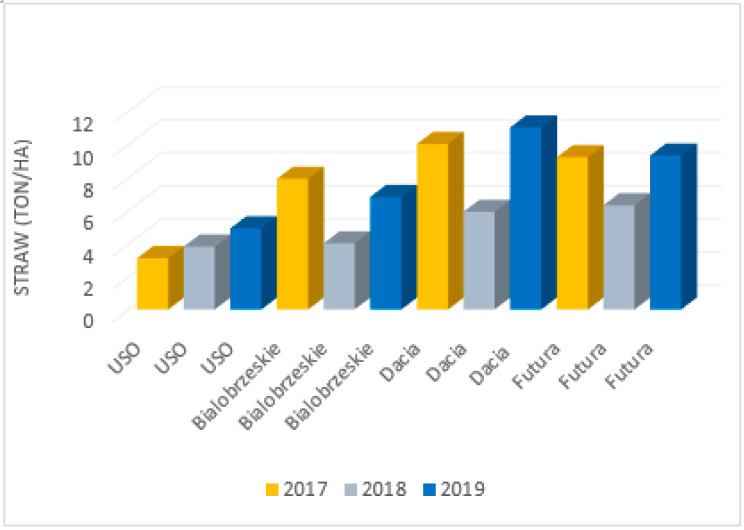
Mean yield of long fibre determined for different type of hemp cultivars: USO 31, Dacia Secuieni, Bialobrzeskie, Futura 75, cultivated for three consecutive years, based on [47].

**Figure 13 materials-15-01901-f013:**
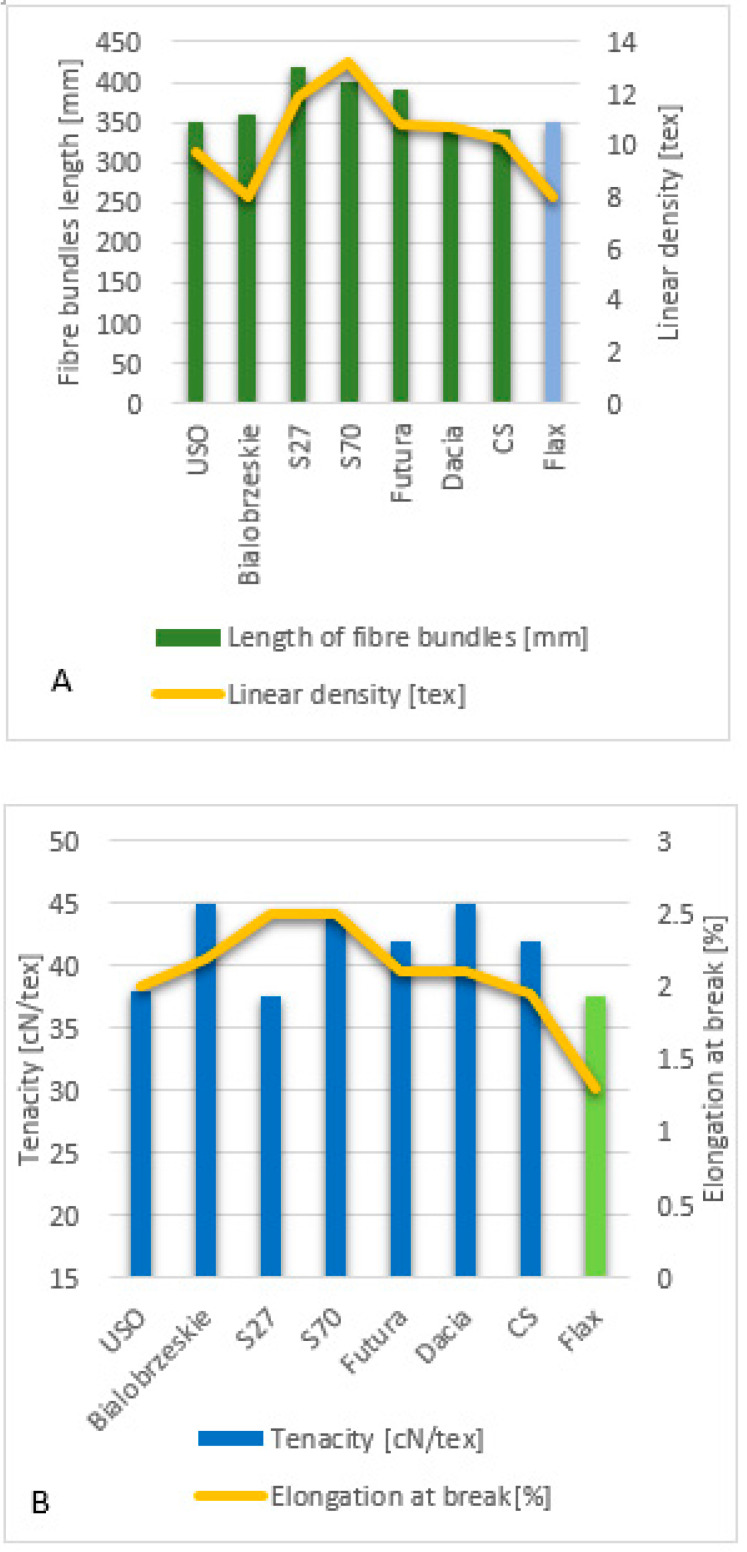
Fibre bundle quality parameters: (**A**)—fibre bundle length vs linear density, (**B**)—elongation at break vs tenacity. Tested hemp variety: USO 31, Bialobrzeskie, Santhica 27, Santhica 70, Futura 75, Dacia Secuieni and Carmagnola Selezionata [47].

**Figure 14 materials-15-01901-f014:**
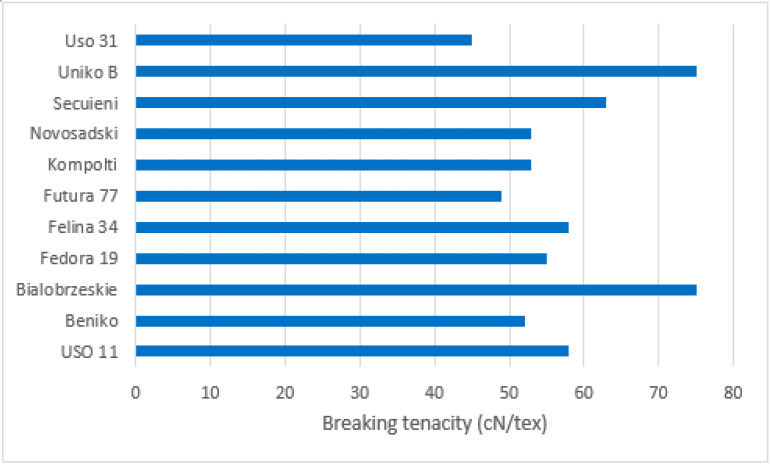
Breaking tenacity of the fibres (cN/tex) measured for 12 fibre samples of each cultivar in 1996, based on [48].

**Figure 15 materials-15-01901-f015:**
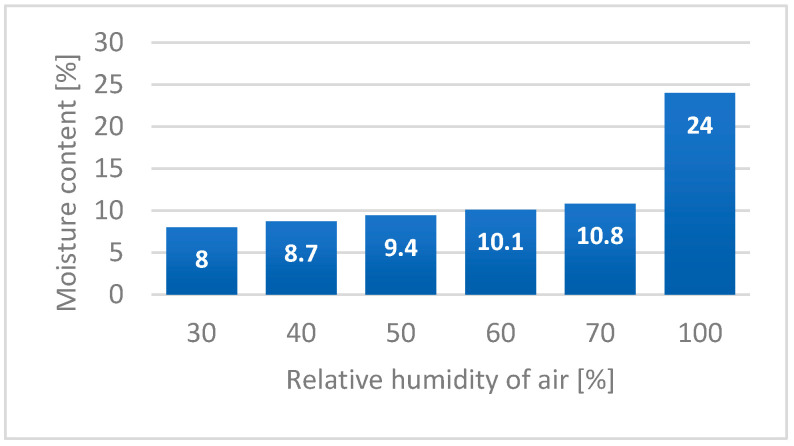
Effect of relative humidity of air on moisture content of hemp fibres based on [51].

**Figure 16 materials-15-01901-f016:**
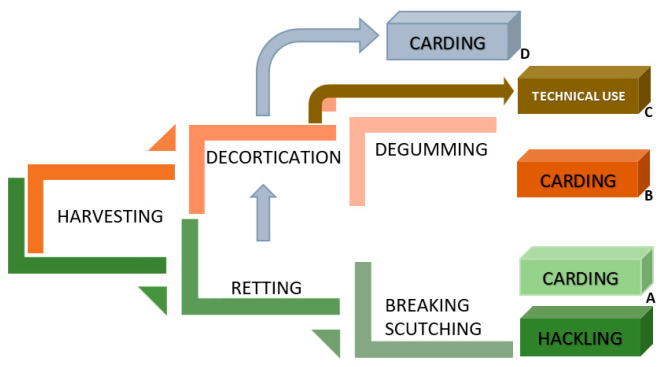
Schematic value chains of hemp fibre extraction for textile purposes. (**A**) (green line)—traditional order of processes including straw retting, (**B**) (orange line)—with use of decortication of straw excluding retting with use of degumming, (**C**) (orange-brown line)—decortication of raw non-retted straw for technical use of the fibres, (**D**) (green-grey line)—with use of decortication following retting aimed at preparation of fibre for cottonisation or woolenisation (*own elaboration*).

**Figure 17 materials-15-01901-f017:**
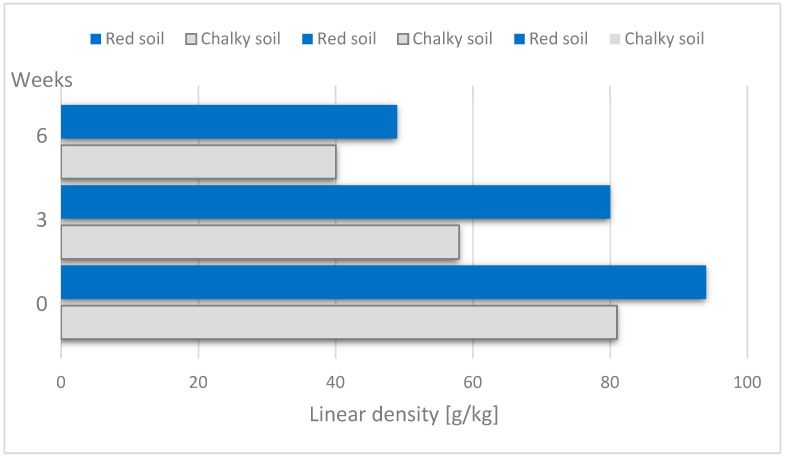
Linear density (g/km) of hemp tow in relation to retting duration, based on [61].

**Figure 18 materials-15-01901-f018:**
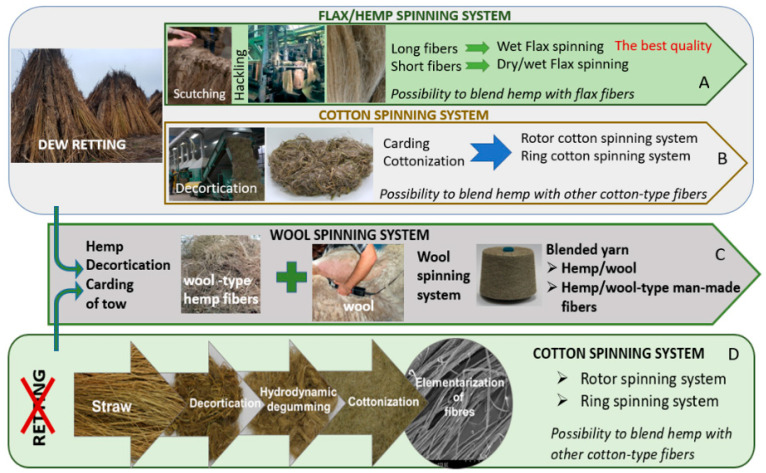
Graphical performance of process chains possible to use for hemp yarn production. **A**—traditional flax/hemp spinning system, **B**—use of cotton spinning system for pre-treated fibres by decortication of retted fibres, **C**—use of wool spinning system, **D**—use of cotton spinning system for pre-treated hemp fibres by degumming of decortication of fibres (*own elaboration*).

**Figure 19 materials-15-01901-f019:**
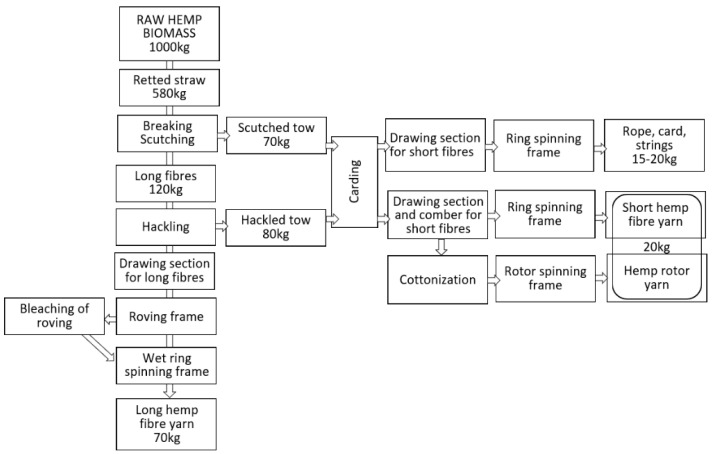
Flow chart of traditional flax/hemp spinning system (*own elaboration*).

**Figure 20 materials-15-01901-f020:**
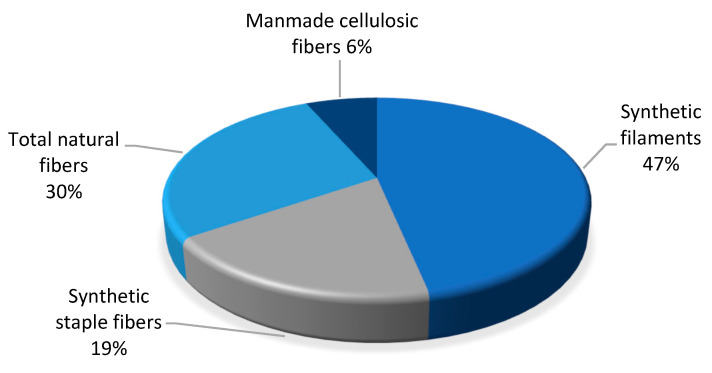
World production of all textile fibres in 2019 based on [115].

**Table 1 materials-15-01901-t001:** Chemical composition of hemp fibres based on the different literature resources ^1^ [43], ^2^ [44], ^3^ [30], ^4^ [31], ^5^ [18], ^6^ [45], ^7^ [38], ^8^ [46].

Cellulose	Hemicelluloses	Pectin	Lignin	Fat/Wax
[%]	[%]	[%]	[%]	[%]
67–75 ^1^	16–18 ^1^	0.8 ^1^	3.0–5.0 ^1^	0.7 ^1^
67 ^2^	16 ^2^	0.8 ^2^	3.3 ^2^	0.7 ^2^
66.0–72.5 ^3^	14–22 ^3^	0.6–3.5 ^3^	2.4–5.5 ^3^	0.2–0.5 ^3^
67–78 ^4^	16–18 ^4^	0.8 ^4^	3.5–5.5 ^4^	0.7 ^4^
68 ^5^	15 ^5^	0.8 ^5^	10 ^5^	
52–58 ^6^	15–18 ^6^	4–10 ^6^	3–6 ^6^	
75 ^7^		9.5 ^7^		
71.41 ^8^			6.59 ^8^	

**Table 2 materials-15-01901-t002:** Diversity of chemical composition of fibres extracted from different hemp varieties with use of water retting: Beniko, Wojko, Tygra and Białobrzeskie. Results are expressed as mean ± standard deviation (SD). Lowercase letters indicate significant differences at *p* ≤ 0.05 according to the Tukay’s HSD test [30].

Variety	Content of:									
	Waxes and Fats (*n* = 3)		Pectin (*n* = 5)		Lignin (*n* = 3)		Cellulose (*n* = 3)		Hemicellulose (*n* = 3)	
	%	±SD	%	±SD	%	±SD	%	±SD	%	±SD
Beniko	0.23 ^a^	0.01	1.47	0.09	2.81 ^a^	0.29	71.31 ^a^	1.32	15.03 ^a^	0.02
Wojko	0.24 ^a^	0.04	0.67 ^a^	0.02	3.02 ^a^	0.31	72.53 ^a^	0.11	16.67	0.24
Tygra	0.25 ^a^	0.04	0.56	0.00	2.78 ^a^	0.28	70.79 ^a^	0.13	15.00 ^a^	0.28
Białobrzeskie	0.34	0.02	0.67	0.02	2.38 ^a^	0.22	72.03 ^a^	0.22	14.37	0.29

**Table 3 materials-15-01901-t003:** Characteristics of hemp fibres based on the different resources of literature ^1^ [44], ^2^ [43], ^3^ [18], ^4^ [46], ^5^ [31], ^6^ [32], ^7^ [42], ^8^ [45].

Linear Density	Density	Microfibril Angle	Diameter of Elementary Fibre	Length of Elementary Fibre	Length of Technical Fibre
tex	g·cm^−3^	◦	μm	mm	mm
^-^	1.07 ^1^	6.2 ^1^	10–51 ^1^	5–55 ^1^	-
^-^	1.52 ^2^	2–6.2 ^2^	10–40 ^2^	-	-
3.0–2.2 ^4^	1.48 ^3^				
0.33 ^5^	1.48–1.49 ^5^		15–30 ^5^	15–25 ^5^	1000–3000 ^5^
1.9–2.8 ^7^			15–50 ^6^		1500–2500 ^6^
	1.43–1.48 ^8^				

**Table 4 materials-15-01901-t004:** Values of mechanical properties of hemp fibres based on data provided by different authors ^1^ [50], ^2^ [43], ^3^ [18], ^4^ [46], ^5^ [19], ^6^ [32], ^7^ [31], ^8^ [42].

Young’s Modulus	Elongation at Break	Strength at Break	Tenacity	Regain
GPa	%	MPa	cN/tex	%
35 ^1^	1.6 ^1^	390 ^1^	43.2–52.2 ^8^	
70 ^2^	1.7 ^2^	920 ^2^		6.2–12 ^2^
70 ^3^	1.6 ^3^	690 ^3^	57 ^3^	
96 ^4^			47–80 ^5^	
	23 ^6^		40–70 ^6^	12 ^6^
	1.5–4.2 ^7^		27–69 ^7^	

**Table 5 materials-15-01901-t005:** Moisture hysteresis of hemp fibres.

Relative Humidity of Air [%]	Moisture Content in Hemp Fibers at 19 °C [%]
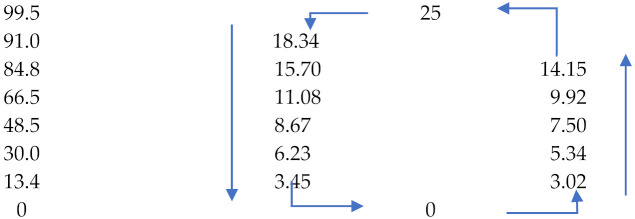	

**Table 6 materials-15-01901-t006:** Fibre moisture content effect on process efficiency and linear density of the fibre.

Fibre Moisture Content [%]	Scutched Fibre Efficiency [%]	Hackled Fibre Efficiency [%]	Mean of Linear Density of Hackled Fibres [tex]	Mean of Linear Density of Scutched Fibres [tex]
8–9	9.3–10.2	67.5–61.8	38.76–40.6	47.62–48.08
10–11	10.9–12.1	70.0–72.1	40.0–40.16	46.95–45.87
11–12	10.9–11.7	67.9–71.9	37.45–41.32	47.62–47.85
14–15	9.8–10.5	63.3–67.4	38.46–38.91	47.4–47.62

**Table 7 materials-15-01901-t007:** Fibre content by weight in stalk based on [61].

	Phloem Content in Dry Straw [%]	Fibres Content in Phloem [%]	Fibres Content in Dry Straw [%]
Hemp	22–32	46–49	10–15

**Table 8 materials-15-01901-t008:** Chemical composition of hemp fibre of the Bialobrzeskie variety after different methods of extraction: dew retting, decortication, osmotically degumming and water retting. Results are expressed as mean ± standard deviation (SD). Lowercase letters (a,b) indicate significant differences at *p* ≤ 0.05 according to the Tukay’s HSD test [30].

Degumming Method	Variety	Content of:									
		Waxes and Fats (*n* = 3)		Pectin (*n* = 5)		Lignin (*n* = 3)		Celullose (*n* = 3)		Hemicelullose (*n* = 3)	
		%	±SD	%	±SD	%	±SD	%	±SD	%	±SD
Decortication		0.47 ^a,b^	0.02	2.00	0.09	5.55	0.17	66.02 ^a^	0.46	21.25	0.05
Dew retting		0.56 ^a^	0.14	3.68	0.19	4.31 ^a^	0.04	66.16 ^a^	0.48	21.72	0.12
Water retting	Białobrzeskie	0.34 ^b^	0.02	0.67	0.02	2.38	0.22	72.03	0.22	14.37	0.29
Osmotic degumming		0.44 ^a,b^	0.04	2.82	0.22	4.03 ^a^	0.09	67.81	0.52	16.29	0.03

## Data Availability

Data available in a publicly accessible repository.

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
