# Peer review of "Hemp Fibre Properties and Processing Target Textile: A Review"

_materials, 2022, doi:10.3390/ma15051901_

Round 1

Reviewer 1 Report

The paper “Cannabis sativa L., industrial hemp fibre properties and processing - opportunities and challenges: a review” is well written review which focus on the interesting area of hemp textiles which meets many challenges. Firstly, are presented the hemp cultivation importance along with the characteristics and properties of hemp fibre. Then the hemp fibre production of the straw and the methods for the production of hemp yarn are fully described. All the used and proposed methods are focused on the production of best quality hemp yarn and end product in the increasing market of natural fibers.

It will be interesting, if included in the paper, some economic aspects of the methods described in figure 18 in order to be obvious the production cost of each method and how possible for an established cotton ginning unit to include cottonization and produce hemp/cotton fibers.

Also the following minor issues has to be taken under consideration

Line 235. This result is also supported under Mediterranean conditions of north Greece, in the below paper.

Tsaliki, E.; Kalivas, A.; Jankauskiene, Z.; Irakli, M.; Cook, C.; Grigoriadis, I.; Panoras, I.; Vasilakoglou, I.; Dhima, K. Fibre and Seed Productivity of Industrial Hemp (Cannabis sativa L.) Varieties under Mediterranean Conditions. Agronomy 2021, 11, 171. https://doi.org/ 10.3390/agronomy11010171

Lines 332-337 need some reference

Lines 350-374 please add some reference and I think that most of the information in this session is repeated when you describe each way of retting separately.

Lines 700-703 add some reference from these mention in introduction

Author Response

The paper “Cannabis sativa L., industrial hemp fibre properties and processing - opportunities and challenges: a review” is well written review which focus on the interesting area of hemp textiles which meets many challenges. Firstly, are presented the hemp cultivation importance along with the characteristics and properties of hemp fibre. Then the hemp fibre production of the straw and the methods for the production of hemp yarn are fully described. All the used and proposed methods are focused on the production of best quality hemp yarn and end product in the increasing market of natural fibers.

  • It will be interesting, if included in the paper, some economic aspects of the methods described in figure 18 in order to be obvious the production cost of each method and how possible for an established cotton ginning unit to include cottonization and produce hemp/cotton fibers.

Thank you very much for the valuable comments. The comments allow me to improve quality of my manuscript.  Economic aspects of the hemp yarn production are very important, although there are no literature records discussing profitability of all types of the technology because methods C and D are not implemented yet to full industrial scale. However I have added short discussion about economic aspects of hemp production.

Also the following minor issues has to be taken under consideration

  • Line 235. This result is also supported under Mediterranean conditions of north Greece, in the below paper.Tsaliki, E.; Kalivas, A.; Jankauskiene, Z.; Irakli, M.; Cook, C.; Grigoriadis, I.; Panoras, I.; Vasilakoglou, I.; Dhima, K. Fibre and Seed Productivity of Industrial Hemp (Cannabis sativa L.) Varieties under Mediterranean Conditions. Agronomy 2021, 11, 171. https://doi.org/ 10.3390/agronomy11010171

Thank you very much for valuable suggestion. I have checked the suggested article. You are right, this article confirms the cited finding. I will use suggested by you references in my future work. Now it is not possible because Reviewer recommendation to include specified reference to the reviewed manuscript cannot be taken into consideration.

  • Lines 332-337 need some reference

I have added references accordingly:  [8, 31, 49]

  • Lines 350-374 please add some reference and I think that most of the information in this session is repeated when you describe each way of retting separately.

I have shortened this part adequately and added references: [76, 85, 86, 93, 104, 106, 113]

  • Lines 700-703 add some reference from these mention in introduction

I have added references accordingly: [6,7]

Reviewer 2 Report

The article reviewed the application, properties, marketing and developing trend of hem fibers. Plan fibers as a kind of sustainable and biobased resources can be widely used in different industries to make added values.

The processing of fibers by steam-explosion could bring about excellent mechanical properties for helm fiber. For environmentally friendly mechanical treatment for helm fibers, the process technique and equipment are important, and therefore some relevant information should be given.

And the mechanical treatments of helm fiber can make the helm fiber based composite products by injection or extrusion-compression widely used. And the performance of the composite can sinificantly enhanced. Which should be concerned.

Author Response

Thank you very much for the valuable comments. The comments allow me to improve quality of my manuscript. I have provided amendments accordingly.

  • Please remove the repeating sentences from the review.

Thank you, I provided some changes, I have shortened description in sec. 3.

  • Please mention the copyright permission in the figure and table legends.

The figures and tables are available in open access publication. Data available in a publicly accessible repository. All captions of cited figures and tables contain respective references. 

  • Why is hemp preferred over the other fibers?. This should be discussed in detail with supporting documents.

This aspect is discussed in section 5. Challenges and Opportunities, but according to Reviewer suggestions I have additionally developed the statement in this section.

  • There are many typo errors please carefully correct all.

Thank you. I have checked the text with use of special software.

  • Nowadays dew retting is the most common method used for flax and hemp. Why flax? please remove flax from the sentence since this review is on hemp.

I mentioned about flax, because the retting method of both flax and hemp are very similar. But I can remove flax from the sentence according to Reviewer suggestions.

  • Please improve the figure quality. Figures 8, 9, 10 may be removed from the review.

I have improved the figures quality, however I would like to have the figures 8, 9,10 at the manuscript, because the structural formulas of main components of fibres fills up the image of discussed chemical composition.

  • What are the new methodologies used for the extraction of hemp fibers, other than the traditional methods? for example, Ultrasonic Retting is frequently used instead of dew and wet retting.

I have presented five main methods of retting not only traditional ones. In section of traditional dew retting, the new one with use of greenhouse retting and additionally not-well-known frost retting have been discussed. Also in case of water retting, new solutions are presented. New solution in enzymatic/fungal retting and physical retting are also discussed. Ultrasonic as a physical method is not used separately, ultrasound are applied in water bath, I mentioned about it in line 601. 

  • What is the current market of hemp? What are the new directions? What are the limitations and strength? The challenges and opportunities should be revised.

The market of hemp is presented in fig. 1, 3, 4 and discussed in introduction and in the section 5. I have developed the section 5. Challenges and opportunities.

  • Some potential applications of hemp are mentioned. Please discuss it all before the challenges and opportunities

Thank you very much for your suggestion. Presentation of potential of hemp use is needed in introduction section to confirm huge multidirectional world interest in hemp-based raw materials and to provide arguments why the hemp is a subject of my work and paper. I would like to keep the structure of the manuscript in the current form.

Reviewer 3 Report

Comments.
Please remove the repeating sentences from the review.
Please mention the copyright permission in the figure and table legends.
Why is hemp preferred over the other fibers?. This should be discussed in detail with supporting documents.
There are many typo errors please carefully correct all.
Nowadays dew retting is the most common method used for flax and hemp. Why flax? please remove flax from the sentence since this review is on hemp.
Please improve the figure quality. Figures 8, 9, 10 may be removed from the review.
What are the new methodologies used for the extraction of hemp fibers, other than the traditional methods? for example, Ultrasonic Retting is frequently used instead of dew and wet retting.
What is the current market of hemp? What are the new directions? What are the limitations and strength? The challenges and opportunities should be revised.
Some potential applications of hemp are mentioned. Please discuss it all before the challenges and opportunities

Author Response

  • The article deals with hemp Fiber processing. The topic is suitable for publication in the journal. However, improvement is requiered. A section dealing with epoxidation of hemos Fiber and fabrication of hemp bionanocomposites should be added. References should be included such as Colomur-Romero Materials 2020, 13, 5720; Shuttleworth J Phys Chem B 2017, 121, 2454.

Thank you very much for your valuable comments. I would like to explain, that the review  discusses hemp fibre from the textile application view point. Such approach to hemp fibre has forced focussing on fibre extraction  suitable for spinning to textile needs. The review does not cover composite materials. Bio-composites or nano-bio-composites reinforced with hemp are out of scope of my current manuscript. I highly appreciate your suggestion to add the specified by you reference, I will use it in my future work on natural fibre based bio-composites.

Reviewer recommendation to use specific reference to the reviewed manuscript should not be taken into consideration.

  • Conclusions should be extended and the future perspectives should be added incorporatinf the author viewpoint.

Thanks for your suggestions, I have improved the conclusion.

The topic is novel and relevant to the journal. To the best of my knoweledge, there are not previous reviews on that topic. Thus, it provides a good contribution to the field. As indicated in my report, conclusions are too short and should be extended including future perspectives and the author opinion.  The other questions do not apply since It is a review article and there are not experiments. 

Reviewer 4 Report

The article deals with hemp Fiber processing. The topic is suitable for publication in the journal. However, improvement is requiered. A section dealing with epoxidation of hemos Fiber and fabrication of hemp bionanocomposites should be added. References should be included such as Colomur-Romero Materials 2020, 13, 5720; Shuttleworth J Phys Chem B 2017, 121, 2454.

Conclusions should be extended and the future perspectives should be added incorporatinf the author viewpoint.

The topic is novel and relevant to the journal. To the best of my knoweledge, there are not previous reviews on that topic. Thus, it provides a good contribution to the field. As indicated in my report, conclusions are too short and should be extended including future perspectives and the author opinion.  The other questions do not apply since It is a review article and there are not experiments.  

Author Response

Reviewer 2

The article reviewed the application, properties, marketing and developing trend of hem fibers. Plan fibers as a kind of sustainable and biobased resources can be widely used in different industries to make added values.

  • The processing of fibers by steam-explosion could bring about excellent mechanical properties for helm fiber. For environmentally friendly mechanical treatment for helm fibers, the process technique and equipment are important, and therefore some relevant information should be given.

Thank you very much for the valuable comments. Problem of lack of machines dedicated to hemp processing has been indicated by me several times in the manuscript. However I made some amendments in section 5 to show that machines deficiency is the main challenge for the hemp textile sector.

  • And the mechanical treatments of helm fiber can make the helm fiber based composite products by injection or extrusion-compression widely used. And the performance of the composite can sinificantly enhanced. Which should be concerned.

Thank you very much for the comments. Use of hemp as reinforcement of composites is very important direction from the bioeconomy view point. However the hemp based composites are very wide area of research, which needs separate manuscript preparation. The multiperspective  review dedicated to composite materials must cover huge diversity of methods of hemp fibres preparation and modification to improve fibre adhesion to polymer matrices, reduce porosity, improve mechanical properties as well as different methods of composites formation including final application, as well as aspects of bio- and nano-bio-composites. Such wide composite research area cannot be included to manuscript dedicated to hemp fibres target textile.  

Round 2

Reviewer 3 Report

The authors have corrected the manuscript. I recommend the manuscript for publication in Materials.

Reviewer 4 Report

The article has been improved. Although not of the suggestions of this reviewer have been taking into account I believe It is now suitable for publication in the journal